# Plasma Synthesis of Advanced Metal Oxide Nanoparticles and Their Applications as Transparent Conducting Oxide Thin Films

**DOI:** 10.3390/molecules26051456

**Published:** 2021-03-07

**Authors:** Hong Yong Sohn, Arun Murali

**Affiliations:** Department of Metallurgical Engineering, University of Utah, Salt Lake City, UT 84112, USA; arunmurali2493@gmail.com

**Keywords:** chemical vapor synthesis, hydrogen-sensing, indium-doped zinc oxide, indium tin oxide, optoelectronics, photocatalysis, plasma, tin-doped zinc oxide, zinc oxide, aluminum-doped zinc oxide

## Abstract

This article reviews and summarizes work recently performed in this laboratory on the synthesis of advanced transparent conducting oxide nanopowders by the use of plasma. The nanopowders thus synthesized include indium tin oxide (ITO), zinc oxide (ZnO) and tin-doped zinc oxide (TZO), aluminum-doped zinc oxide (AZO), and indium-doped zinc oxide (IZO). These oxides have excellent transparent conducting properties, among other useful characteristics. ZnO and TZO also has photocatalytic properties. The synthesis of these materials started with the selection of the suitable precursors, which were injected into a non-transferred thermal plasma and vaporized followed by vapor-phase reactions to form nanosized oxide particles. The products were analyzed by the use of various advanced instrumental analysis techniques, and their useful properties were tested by different appropriate methods. The thermal plasma process showed a considerable potential as an efficient technique for synthesizing oxide nanopowders. This process is also suitable for large scale production of nano-sized powders owing to the availability of high temperatures for volatilizing reactants rapidly, followed by vapor phase reactions and rapid quenching to yield nano-sized powder.

## 1. Introduction

### 1.1. Overview of Transparent Conducting Oxides (TCO’s)

A TCO is a wide band-gap semiconductor that has high concentration of free electrons in its conduction band. These arise either from defects in the material or from extrinsic dopants, the impurity levels of which lie near the conduction band edge. The high-electron-carrier concentration causes absorption of electromagnetic radiation in both the visible and infrared portions of the spectrum for the present purposes, with the former being more important. As a TCO must necessarily represent a compromise between electrical conductivity and optical transmittance, a careful balance between the properties is required [1].

Generally, TCOs are metal oxides with high optical transmittance and high electrical conductivity [2]. They are also referred to as wide-bandgap oxide semiconductors (band gap > 3.2eV). These materials have high optical transmission at visible wavelengths (400–700 nm) and electrical conductivity close to that of metals, which is often induced by doping with other elements. They also reflect the near infrared and infrared wavelengths. Since the bandgaps of these materials lie in the ultraviolet wavelength region they hardly absorb visible light; thus, they appear to be transparent to the human eye [3].

Transparent conducting oxides (TCOs) are used in a wide range of applications, including low-e windows, transparent contacts for solar cells, optoelectronic devices, flat panel displays, liquid crystal devices, touch screens, electromagnetic interference (EMI) shielding, and automobile window deicing and defogging. Low cost, high durability and being non-toxic make ZnO an attractive alternative to the commonly used ITO. ZnO has a direct and wide band gap in the near-UV spectral region and a large free-exciton binding energy so that excitonic emission processes can persist at or even above room temperature. One of the key challenges in developing ZnO-based TCOs is investigating the best metal dopants and the optimal dopant contents in order to achieve the highest electrical conductivity. Unlike in SnO_2_ and In_2_O_3_-based TCOs, efficient doping of group III elements into the ZnO structure could decrease the resistivity significantly, potentially realizing a future low cost TCO for electronic and optoelectronic applications [4,5].

### 1.2. Opto-Electrical Properties of TCOs

For electrical conduction to occur within a semiconductor material, ground state electrons must be excited from the valence band to the conduction band minimum (CBM). A wider band gap requires a higher-energy photon. Therefore, widening the band gap (i.e., Eg > 3.0 eV) in a material permits transparency to the visible portion of the spectrum by placing a greater separation between the valance band maximum (VBM) and CBM of the material, thus decreasing the probability of exciting an electron into conduction [6]. TCOs have been developed by doping materials in order to facilitate the charge carrier generation within the structure. In the description of the band model as shown in Figure 1, there is an important difference between the fundamental band gap (i.e., the energy separation of the E_vb_ and E_cb_; an intrinsic property of the material) and the optical band gap (an extrinsic property), which corresponds to the lowest energy allowed for an optical transition. 

The optical band gap determines the transparency of a material which is important in TCO applications. In order to achieve n-type conducting properties, electrons are injected from a nearby defect donor level directly into the conduction band. The point defects in a metal oxide crystal, such as oxygen vacancies, proton or metal interstitials and certain substitutional defects, effectively create an excess of electrons close to the defect site in n-type TCOs. If there is sufficient orbital overlap, it permits delocalization of electrons from the defect sites such that electronic states at the CBM become filled or in other words Fermi level shifts above the CBM. This leads to an effect known as the Moss–Burstein shift, which effectively widens the optical band gap [3].
E_g_ = E_CBM_ − E_VBM_(1)
E^opt^g = E^MB^g + Eg = E_F_ − E_VBM_(2)

Since the Moss–Burstein shift is:E^MB^g = E_F_ − E_CBM_(3)
where E_g_ is the fundamental energy gap separating the VBM and CBM, E^opt^g is the optical band gap corresponding to the smallest allowed optical transition from the VB to the CB, E^MB^g is the Moss–Burstein shift and E_F_ is the Fermi level. Thus, lattice defects in TCOs can simultaneously promote both electrical conductivity and optical transparency. Apart from the Moss–Burstein shift, the fundamental band gap is tapered due to the band gap narrowing effect which led by exchange interactions in the free-electron gas and electrostatic interactions between free electrons and ionized impurities [7]. The optical band gap is a key aspect in designing a TCO. However, the CBM depth or electron affinity (EA), in other words the difference between valence energy and CBM which affects the “dopability” of the TCO, is also equally important in determining the conducting properties. A higher value of EA indicates greater ease of introducing charge carriers, i.e., a greater dopability [8]. A large separation (Eg > 3.0eV) between the Fermi level in the conduction band and the next electronic energy level (CBM+1) helps to prevent excitation of electrons to higher states within the conduction band, which prevents undesirable optical absorption [9].

### 1.3. Plasma-Assisted Chemical Vapor Synthesis

Chemical vapor synthesis involves introducing vapor phase precursors into a heated reactor and allowing vapor phase nucleation to occur in the reactor rather than depositing the product as a thin film. The precursors can be solid or liquid in nature but are introduced into the reactor as a vapor by sublimation or evaporation [10]. Chemical vapor synthesis assisted by thermal plasma involves the use of a plasma flame as the heat source and types of plasma torches are illustrated in Figure 2 [11]. 

Plasma flames can facilitate vapor phase reactions by providing sufficient energy for vaporizing precursors and subsequent chemical reactions. The temperature of plasma flame generated is high enough to decompose even reactants of high vaporization temperatures into atoms and radicals, which can then react and condense to form nanosized particles when cooled by mixing with cool gas or expansion through a nozzle as shown in Figure 3. 

Thermal plasma provides a high processing rate and other advantages like good control over size, shape and crystal structure as well as a clean reaction atmosphere that allows high purity products, a high quench rate to form ultra-fine powder, and a wide choice of reactants [12,13]. Compared with other methods it avoids multiple steps like in mechanical milling, sol-gel method and precipitation method and does not require a high liquid volume and surfactants that are involved in a wet chemical process. Precursors like chloride salts are commonly used in wet chemical methods but the presence of chloride ions form hard agglomerates in oxide particles and also difficult to be rinsed from the colloidal hydroxide precipitate [14].

The plasma reactor system used in this work [15] for the synthesis of indium tin oxide (ITO) nanopowder consisted of a downward plasma torch, a cylindrical reactor, a powder feeding system, cooling system, powder collectors, and an off-gas scrubber, as shown in Figure 4. The plasma torch had a copper cathode and a tungsten anode. It was water-cooled and operated at atmospheric pressure. The reactor assembly consisted of a vertical stainless-steel tube, which was water-cooled, and had the dimensions of 15 cm inner diameter and 60 cm length. A graphite cylinder of 7.6 cm inner diameter and 60 cm length was placed inside the stainless-steel tube. A graphite felt filled the gap between the graphite cylinder and the stainless-steel tube. The cooling chamber consisted of two-layer stainless steel to cool the outgoing gases below 150 °C. The precursor was directly fed into the plasma gun using Ar as the carrier gas through a powder feeding system consisting of a test tube filled with the precursor powder, a motor that pushed up the test tube at a constant rate, a carrier gas line that carried the fluidized particles from the top of the particle bed in the test tube, and a vibrator. The product was collected on a Teflon coated polyester filter with a pore size of 1 µm. The off-gas scrubber used a 5% NaOH solution. More details about the experimental set-up can be found in previous publications [13,16].

## 2. Synthesis of Indium Tin Oxide (ITO) Nanopowder

ITO is an n-type transparent conducting oxide (TCO) formed by substitutional doping of In_2_O_3_ in which In^3+^ sites are substituted with Sn^4+^ sites. It combines high conductivity with optical transparency in the visible region. It is mainly used to make transparent conducting coatings in electronic displays, heat-reflective coatings for architectural, automotive and light bulb glass, electrochromic windows, solar cells and gas sensors [17]. Transparent conducting oxides (TCO) in general are of great scientific and commercial importance because of their high optical transparency in the visible region and excellent conductivity. TCO’s are commonly used in transparent transistors, gas-sensing devices, light emitting diodes and electro-optical devices [18].

Indium and tin alkoxides have been used for the synthesis of ITO, but these precursors are expensive, water sensitive and difficult to control the synthesis process [19]. Thin films of ITO on glass substrates have been made by a variety of techniques such as chemical vapor deposition [20], spray pyrolysis [21], vacuum evaporation [22], magnetron sputtering [23] and electron-beam evaporation for transparent and conductive electrodes, albeit with inherently costly and time-consuming processes. It has been reported that nanoparticles can be handled efficiently and the loss of raw materials can be minimized [24]. 

Much of the previously published reports extensively deal with nanostructured ZnO-based sensor for gas-sensing applications. Doping with noble metals (such as Pt, Pd) or other alternatives (such as Al, Cu) improved the selectivity and sensitivity of ZnO-based sensor and was used to detect several gases. Recent research has shown transparent and conducting ITO films offer new advantages in the design of metal-oxide based sensor [25,26]. 

The precursors used in this work were 1) indium nitrate (In(NO_3_)_3_·5H_2_O, Alfa Aesar, Haverhill, MA, USA) and 2) tin nitrate(Sn(NO_3_)_4_, American Elements, Los Angeles, CA, USA). Each precursor was ground by mortar and pestle and sieved until the final size became approximately 50 µm. The milled precursors were dried in a vacuum oven at 50 °C and. Mixed using a vibrating mixer. The amount of tin nitrate was varied to obtain In/Sn atomic ratios of 95:5, 90:10 and 85:15 designated as ITO1, ITO2 and ITO3, respectively.

### 2.1. Powder Characterization

The synthesized powders were characterized through the X-ray diffraction technique (Rigaku D/Max-2200V) for its structural analysis. The surface morphology of the powder was investigated by High Resolution Field Emission Scanning Electron Microscope (Hitachi S-4800) attached with Energy Dispersive Spectrophotometer (EDS) system. XPS (Kratos Axis Ultra DLD) was utilized to analyze the chemical state of ITO nanopowder. Raman scattering spectra was measured using micro-Raman spectroscopy (WITec Alpha SNOM) using a He-Ne laser as the excitation source with holographic grating of 1800 grooves/mm. 

#### 2.1.1. XRD Analysis

The XRD patterns of ITO nanopowder showed the cubic bixbyite structure of In_2_O_3_. The grain size increased from 18 nm to 36 nm as the plasma power increased from 10 kW to 30 kW. The crystallite size was calculated from Debye-Scherrer equation [27]:D = Kλ/(β cos θ)(4)
where β = √(β^2^_(FWHM)_ − β^2^_0_) is the peak broadening after subtracting the instrumental broadening effect, β_(FWHM)_ is the full width at half maximum and β_0_ is the correction factor (0.005 rad). 

At the highest plasma power of 30 kW, SnO phase was observed along with the main In_2_O_3_ peaks, since at higher temperature the SnO phase is thermodynamically more stable than the SnO_2_ phase. Raman Spectroscopy yielded results in agreement with the XRD results, by confirming the existence of cubic bixbyite structure of In_2_O_3_ [15]. 

#### 2.1.2. X-ray Photoelectron Spectroscopy (XPS)

The peaks located at 444.4 eV and 452.1 eV in Figure 5a correspond to the In 3d_5/2_ and In 3d_3/2_ states, respectively, which represent the In^3+^ bonding states from In_2_O_3_. Figure 5b shows the peaks of Sn 3d_5/2_ and Sn 3d_3/2_ at 486.8 eV and 495.2 eV, respectively, which indicate the Sn^4+^ bonding states. The Sn 3d peaks are sharp without peak splitting, which confirms the absence of Sn^2+^ bonding state [28]. The XPS results show an asymmetric shape of O_1s_ spectra, Figure 5c. The main peak centered at 529.92 eV is associated with O^2−^ species in the cubic bixbyite structure of In_2_O_3_. The shoulder at higher binding energy contains the contributions from two components that are oxygen vacancies or defects and chemisorbed oxygen species [29]. 

#### 2.1.3. Scanning Electron Microscopy—Energy Dispersive Spectroscopy (SEM-EDS)

Figure 5a shows representative SEM micrographs of ITO synthesized at 10 kW and 20 kW. The particles have hexagonal and nearly spherical morphology. The particle size from the SEM micrograph in the range of 15–25 nm is in agreement with the result from the XRD analysis. Figure 6b shows the EDS spectrum of ITO synthesized at 10 kW. It indicated the presence of indium and tin elements where the In Lα_1_ and Sn Lα_1_ peaks appeared at 3.3 keV and 3.5 keV, respectively. The obtained EDS results for all samples indicated that the elemental distribution of In and Sn is uniform and the Sn/In (atomic %) in the product is close to the initially designed composition of ITO nanopowder.

### 2.2. Thin Film Characterization Results

To obtain uniform coating, the glass slides were first washed with deionized water and then sonicated with acetone for 10 min at 50 °C and washed again with deionized water to remove any trace of acetone on the glass slide. The glass slides were dried in a drying oven at 80 °C prior to the coating operation. ITO nanopowders with different Sn amounts were then mechanically dispersed in ethanol with ammonium polyacrylic acid added as a dispersion agent to obtain an ITO sol. The suspension had a blue color and showed no settling at a solid content of 10 wt %. The suspension was spread on a 2.5 cm × 2.5 cm borosilicate glass substrate and rotated at 2000 rpm for ITO sol coating. The prepared coating was thermally densified at 500 °C in Ar atmosphere for 1 h.

The ITO films were then analyzed using a Rigaku D/Max-2200V X-ray diffractometer with Cu Kα radiation (λ = 1.5406 Å) from 10.00° to 80.00° at a rate of 0.02°/second. The sheet Resistance of the thin film was measured by the 4-probe technique. Hall effect measurements were carried out using the four-terminal method to minimize the Schottky contacts. The optical properties were recorded using a UV-Vis-NIR spectrophotometer (Shimadzu UV-3600). 

#### 2.2.1. XRD Analysis

X-ray diffraction showed the cubic bixbyite structure of In_2_O_3_.

#### 2.2.2. Electrical properties

The I-V characteristics of all ITO thin films indicated the ohmic behavior by yielding a linear behavior. The thicknesses of all the films measured from the SEM cross-sectional image was 350 ± 5 nm. The decrease in resistivity with Sn doping is due to the replacement of In^3+^ ions by Sn^4+^ ions leading to an increase in the density of charge carriers as more number of free electrons are available for conduction, and is shown in Table 2. 

### 2.3. Optical Properties

Optical transmission spectra of ITO films are recorded in the wavelength region of 200–800 nm. Achieving an ITO thin film for transparent electrode with low resistivity and high transparency has always been the goal of the process. Figure 6a shows the transmission spectra of ITO films. The maximum transmittance ranges from 85% to 75% as the Sn content increases due to an increase in the scattering centers and increasing defects in grain boundaries.

The optical absorption coefficient α of a direct band gap semiconductor near the band edge, for a photon energy *hν* greater than the band-gap energy *E_g_* of the semiconductor, is given by the relation [30]
(5)$$(\alpha h\nu )=A{\left(h\nu -E_{g}\right)}^{\frac{1}{N}}$$
where *h* is Planck’s constant and *ν* is the frequency of the incident photon. The constant N depends on the nature of electronic transition. In the case of ITO films, N is equal to 2 for direct allowed transition.

The Tauc plot of (*αhν*)^2^ versus energy hν for all the ITO films are shown in Figure 7b. The band gap energy was obtained by extrapolating the linear plot of the Tauc plot curves to the intercept with the energy axis (at *αhν* = 0). It was observed that the band gap increased from 3.6 eV to 3.7 eV with increasing Sn concentration, as in ITO2 film. The blue shift exhibited on increasing the Sn concentration is associated with the Moss Burstein (BM) effect. According to this well-known effect, the conduction band of the degenerate semiconductor is filled with high carrier concentration and the lowest valence energy states are blocked, leading to the lifting of the Fermi level into the conduction band and widening of the optical band gap [31]. At a high doping concentration, that is in ITO3, the decrease in transmittance is due to excess Sn atoms segregating to grain boundaries, which can be associated with an amorphous structure of the ITO3 film. The decrease in band gap in ITO3 to 3.65 eV can be again explained by the decrease in carrier concentration, which causes a red shift, indicating that the variation in band gap is dependent on the change in the electron concentration.

### 2.4. Gas Sensitivity

After depositing the ITO film on a glass substrate, the device was fabricated as a resistive-type gas sensor. The substrate was 2.5 cm × 2.5 cm in size. An ITO film of thickness 300 nm deposited by the spin-coating process was used to fabricate the device. The sensor was then placed in an alumina tube inserted coaxially inside a tubular furnace so as to adjust and optimize the temperature of the sensor. For response measurements, a known volume of gas was flowed inside the alumina tube and subsequently a decrease in the electrical resistance of the film was observed. The gas concentration was controlled by mass flow controller and the electrical resistance before and after exposure to H_2_ gas was measured using a Keithley 2000 DMM. The film was tested for the operating temperature in the range of 100 °C to 500 °C in the steps of 50 °C and the gas concentration was varied from 50 ppm to 600 ppm. The resistance of the sensor in H_2_ gas was measured as a function of time up to its saturation in response and then air was flowed into the tube. The resistance in air was measured until the recovery of sensor resistance to its original value in air.

The ITO film exhibited good sensitivity to H_2_ gas, and the sensitivity increased with increases in gas concentration and temperature and reached maximum with 400 ppm of H_2_ gas at an operating temperature of 350 °C.

## 3. Synthesis of Zinc Oxide and Tin-Doped Zinc Oxide (TZO) Nanopowders

Transparent conducting zinc oxide and tin-doped zinc oxide (TZO) nanopowders were synthesized for the first time using a novel plasma-assisted chemical vapor synthesis route [32]. Zinc oxide (ZnO) is a II-VI group compound semiconductor which crystallizes in wurtzite structure belongs to the space group P6_3_mc and has properties of high optical transparency, non-toxicity, good UV trapping, etc. which are essential properties of optoelectronic and piezo electronic materials due to its large band gap of 3.37 eV and exciton binding energy of 60 meV [33]. However, ZnO films have poor conductivity and doping with various dopants is usually necessary to improve the conductivity for as use as TCO film. Replacing Zn^2+^ ions with higher valence ions such as Al^3+^ and Sn^4+^ ions brings about significant changes in electrical and optical properties [34]. At present, tin-doped indium oxide (indium tin oxide or ITO) is the most commonly used TCOs, but due to concerns of the supply of world indium reserves and cost of indium, there has been an increasing interest in alternatives [35].

Tin-doped zinc oxide (TZO) is an important alternative to ITO and is widely used as transparent electrode in various kind of devices. When Sn was added into ZnO for doping, Sn^4+^ substitutes Zn^2+^ sites in the ZnO crystal structure because of its smaller ionic radius (0.069 nm) than Zn^2+^ (0.074 nm) which results in the addition of two more free electrons and thereby improving the electrical conductivity [36]. The electrical conductivity, transparency, thermal stability, and durability make this material interesting and attractive [37,38]. 

Different synthesis techniques of TZO nanoparticles have been reported. Wu et al. [39] fabricated Sn-doped ZnO nanorods by hydrothermal treatment and used methyl orange as the probe molecule to evaluate its photocatalytic activity. Wang et al. [19] prepared tin-doped zinc oxide nanoparticles in organic solution, with metal acetylacetonate as the precursor and oleyl amine as the solvent. Javid et al. [40] synthesized TZO nanoparticles by chemical solution method using zinc nitrate and NaOH as precursors. Junlabhut et al. [41] synthesized TZO nanopowders by co-precipitation method with various Sn additives from 0–50 wt %. Verma et al. [42] reported the structure-property relationship in undoped and Sn-doped ZnO nanostructured materials synthesized by co-precipitation. Li et al. [43] used the substrate of p-type Si (100) for synthesizing TZO nanowires involving a vapor-liquid-solid growth process. 

In this laboratory, zinc oxide and tin-doped zinc oxide nanopowders were synthesized by the plasma process and tested for its photocatalytic property in the degradation of methylene blue. The obtained nanopowders were also used to fabricate ZnO and TZO films to investigate the influence of Sn doping on their structural, optical, electrical and photocurrent properties. The high figure of merit and resistivity values lower than reported values on TZO films [44,45] indicated the suitability of plasma-synthesized ZnO and TZO as transparent electrical contacts in optoelectronic devices.

The precursors used for synthesizing zinc oxide and tin-doped zinc oxide in this work were as follows: (1) zinc nitrate powder for zinc oxide, and (2) a mixture of zinc nitrate and tin nitrate powders for tin-doped zinc oxide. The amount of tin nitrate was altered to achieve 3 and 5 atomic percent Sn designated as TZO1 and TZO2, respectively.

### 3.1. Powder Characterization

#### 3.1.1. X-ray Diffraction (XRD)

The XRD patterns of ZnO, TZO1 and TZO2 all showed the peaks corresponding to the hexagonal wurtzite structure of ZnO with good crystallinity [32]. No SnO_2_ or other Sn phases were detected in TZO1 sample but in TZO2 sample, SnO_2_ phase was observed indicating that the Sn content exceeded its maximum solubility in ZnO. The presence of SnO_2_ phase as observed in TZO2 lowers the crystallinity of TZO nanopowder which is evident from the decrease in the intensity of peaks. 

Crystallite size is decreased with Sn concentration at 3 atm % (TZO1), which is due to lattice shrinkage from the variations in the ionic radii of Zn^2+^ and Sn^4+^ ions. However, an increase in the crystallite size is observed in TZO2 (5 atm %), which is due to an excess segregation of Sn^4+^ ions to its grain boundaries. Since the (101) reflection is the highest in all the samples, we can conclude the particles are orientated mostly in the direction. In TZO2, excess Sn^4+^ serves as an inhibitor for the growth of particles in the direction. The crystallite size decreased on increasing the doping amount to 3 atm % as in TZO1 and a simultaneous decrease in FWHM is also observed as shown in Table 1.

#### 3.1.2. Raman Spectroscopy

ZnO has a wurtzite structure and the symmetry of the ZnO crystal structure lies in the spatial group of C­­^4^_6v_ with two formula units in the primate cell. 

In this study, as shown in Figure 8, a strong and a sharp peak at 437 cm^−1^ is attributed to the high frequency of E_2_(high) mode and is the characteristic feature of hexagonal wurtzite structure of ZnO [46]. ZnO and TZO1 exhibited high intense Raman peaks suggesting high crystal quality whereas for TZO2 the intensity of E_2_(high) mode decreased indicating that the crystallinity of TZO2 nanopowder is deteriorated at a higher doping amount and this observation corroborates with the XRD data. 

#### 3.1.3. X-ray Photoelectron Spectroscopy (XPS) and Photoluminescence Spectroscopy (PL)

XPS analysis was performed to ascertain the concentration of Sn and to ascertain the valence of Sn in TZO1 and TZO2. The difference in binding energy of 23.10 eV between Zn 2p_3/2_ and Zn 2p_1/2_ and the binding energy position indicates that Zn in ZnO, TZO1 and TZO2 nanopowders exist in +2 oxidized state. To ascertain the valence state of Sn, Sn 3d spectra was also obtained from the samples. The Sn 3d_5/2_ and Sn 3d_3/2_ peaks are located at 486.6 eV and 495.2 eV, respectively. The obtained Sn 3d_3/2_ signal was intense due to the Auger ZnL_3_M_45_M_45_ transition [47], as a result it was difficult to compare the spectral parameters of Sn 3d_3/2_ with those of the reference standard. Binding energy position of Sn 3d_5/2_ at 486.6 eV indicated that Sn is incorporated in the form of Sn^4+^ state [48]. This further gives support to the Sn^4+^ ions substituting for Zn^2+^ sites. Zn 2p spectra are located at 1021.42 eV and 1044.34 eV, as shown in Figure 9a, which corresponds to Zn 2p_3/2_ and Zn 2p_1/2_, respectively. Intensity of Sn 3d_5/2_ increased with doping amount of Sn as shown in Figure 9b. The PL spectra of undoped and Sn-doped ZnO nanopowders are shown in Figure 10. All of the spectra showed a strong UV emission peak and a green emission peak (deep-level emission) in the visible region. The weak green band edge emission in the PL spectrum for undoped ZnO indicates low concentration of oxygen vacancies present. This further confirms that in TZO, Sn atoms exist at substitutional sites and share the oxygen with Zn atoms and hence increases the concentration of oxygen vacancies enhances the intensity of green band emission.

#### 3.1.4. Scanning Electron Microscopy—Energy Dispersive Spectroscopy (SEM-EDS)

SEM micrographs of TZO1 particles are show in Figure 11a. Spherical shaped morphology with no agglomeration was seen for all compositions of ZnO and TZO particles. EDS spectra for TZO1 nanoparticles are shown in Figure 11b, which confirmed that the elemental distribution of zinc and tin is uniform and the Sn/Zn (atm. %) composition in the nanoparticles nearly matches with designed compositions. 

### 3.2. Measurement of Photocatalytic Activity

Photocatalytic degradation of methylene blue by ZnO and TZO with various doping concentrations was carried out in 1-liter volume reaction vessel, as shown by a schematic sketch in Figure 12. The quartz immersion well is double-walled and consisted of an inner diameter tube through which cooling water was flown. Cooling water was circulated to control the solution temperature. 450 W mercury vapor lamp was inserted vertically in the immersion well. Approximately 60% of the radiated energy is in the ultraviolent portion of the spectrum, 35% in the visible region and the balance in the infrared range. 

Dyes are important source of environmental pollution and it is used in many industrial processes, namely cosmetic, textile, and printing [49]. Nearly 15% of the world production of dyes, lost during the process of dying is released as textile effluents [49]. Methylene Blue, which is a cationic dye is primarily used in the textile industry and the discharge of large amounts of effluent is harmful to microbes, aquatic system and human health [50]. The colored water is detrimental to environment since its color blocks the sunlight access to aquatic organism and plants, which in turn diminishes photosynthesis and affects the ecosystem [51]. In recent years, an advanced oxidation process utilizing photocatalyst has gained a significant deal of importance in wastewater treatment because of its low consumption of energy, mild reaction condition and high activity compared to other reported methods [52,53].

Photocatalytic activity of the prepared slurry was evaluated by the degradation of a methylene blue solution. The slurry was prepared with 0.1 g of the photocatalyst dispersed in 500 mL of 85 µM methylene blue (MB) solution. The centrifuged MB solution sample was filtered to remove any particle. The filtrate was then analyzed by using a UV-VIS spectrophotometer (Shimadzu UV-3600) by measuring the light absorption intensity at 664 nm where the maximum absorption intensity was attained. The reaction kinetics of the photocatalysis obeyed the following equation:(6)$$R=\frac{dC}{dt}=-kt$$
where *C* is the concentration of methylene blue at time *t* and *k* is the reaction rate constant.

Integration of Equation (3) gives:
Ln(*C*) = −*kt* + Ln(*C_i_*) (7)
where *C_i_* is the initial concentration of methylene blue. The reaction rate constant (*k*) was calculated from the slope of ln(*C*) vs. time plot. 

#### Photocatalysis Data and Absorption Spectrum Analysis

The photocatalytic activity of ZnO, TZO1 and TZO2 nanopowders were investigated using MB as an indicator under UV light. The results demonstrated that TZO1 is a superior photocatalyst to ZnO. The degradation rate constant for TZO1 increased to 0.0339 min^−1^ from 0.0106 min^−1^ as shown in Figure 13a. The enhanced photocatalytic efficiency of TZO1 is attributed to an enhancement in the specific surface area and also due to the band gap of the nanopowder. Upon doping Sn in ZnO, the BET specific surface area increased from 18.5 m^2^/g to 48.6 m^2^/g. The enhanced S/V ratio facilitated an increase in the number of the active states on the photocatalyst surface and thereby increased the concentration of photo generated carriers [54]. Thus, the relative number of free radicals attacking the dye molecules increases. Another factor influencing the rate of degradation is the band gap. Figure 13b shows TZO1 nanoparticles have maximum absorbance at 395 nm that corresponds to a band gap of 3.16 eV which was redshifted compared to undoped ZnO nanoparticles. ZnO showed a maximum absorbance at 383 nm corresponding to a band gap of 3.22 eV and thereby indicating that TZO nanoparticles have higher absorption that is TZO nanoparticles have photocatalytic activity higher than ZnO nanoparticles. The red shift observed with lower band energy was also observed for other doped ZnO samples [55,56]. 

Higher photocatalytic activity in TZO1 is also attributed to higher oxygen vacancies, revealed from the PL spectra. Oxygen vacancy is an important reason for band gap narrowing, as observe previously [57]. The photocatalytic activity of TZO2 was lower than that of TZO1 due to the fact that at higher doping concentrations, the excess Sn cannot enter the lattice of ZnO crystal or further alter the bandgap and segregates at grain boundaries to form defect clusters. The oxygen vacancy concentration was lower for TZO2 (5 at. % Sn) than for TZO1 (3 at. % Sn) corresponding to a decline in the photocatalytic activity of TZO2 sample.

### 3.3. Thin Films Analysis

#### 3.3.1. X-ray Diffraction (XRD)

All the films exhibited a sharp peak near 34.54° corresponding to the (002) plane of the hexagonal wurtzite structure of ZnO and exhibited a *c*-axis orientation perpendicular to the substrate surface [58]. With the addition of Sn, the crystalline quality of the TZO1 thin film remained the same as the ZnO film. In TZO1, the (002) plane became sharper and narrower than the TZO2 film. The lattice distortion is due to segregation of excess Sn to its non-crystalline regions that act as centers of scattering and thereby reduces the orientation of c- axis [59].

#### 3.3.2. Electrical Properties

Room temperature electrical properties of the thin films were recorded using the four-probe technique. Current vs. voltage curve showed a linear relationship, which demonstrated the ohmic nature of undoped and doped films. The resistivity of TZO1 decreased to 1.4 × 10^−3^ Ω cm from 8.2 × 10^−3^ Ω cm for the undoped ZnO thin film. In TZO2 film, increase in resistivity was observed which is attributed to the segregation of excess Sn dopant at grain boundaries which act as carrier traps [60]. 

The reported minimum resistivity values in the literature [44,45] were much higher than our current study. Hall measurement derived electrical properties of films is shown in Table 2 and it is evident that the mobility reached maximum value in TZO1 for which the resistivity had a minimum value.

#### 3.3.3. Optical Properties

The optical properties of the thin films were determined in the wavelength range of 300 to 800 nm. The transmission spectra of all the films exhibited an average transmission of 80%. All films exhibited a ripple pattern, which revealed a homogeneous surface and good adhesion on glass substrates. The general requirements of a TCO thin film for practical use are a resistivity of <10^−3^ Ω·cm and a transmittance of >80% in visible range. Thickness, substrate, growth temperature, dopant, and their content play a crucial role in structural, electrical, and optical properties of films. The values of the transmission obtained vary from 75% to 85% as reported in the previous reports for tin-doped zinc oxide thin films [44,62,63,64]. For applications as transparent contacts, a film must have a low resistivity and high transmissivity in the region of visible spectrum. The figure of merit (*FOM* = *T*^10^/*Rs*) where *T* is the average transmissivity in the visible region and *Rs* is the sheet resistance is usually calculated as the criterion to determine the performance of transparent conducting oxides and the high figure of merit of 3.2 × 10^−3^ is obtained for TZO1 in this study.

#### 3.3.4. Photocurrent Properties

The photocurrent characteristics of the prepared film indicated that only the TZO1 film showed an enhancement in current response when irradiated with UV light. An increase in photocurrent is due to the photo excitation of electrons from VB into CB. When the films are irradiated with UV light, photon energy is absorbed and since the photon energy of UV light is higher than the optical band gap energy of the films, phot-generated charge carriers are created [65]. 

## 4. Synthesis of Aluminum-Doped Zinc Oxide (AZO) Nanopowders

Transparent conducting oxide aluminum-doped zinc oxide (AZO) nanoparticles were synthesized by the plasma-assisted chemical vapor synthesis route described above using zinc nitrate and aluminum nitrate as the precursors [61,66]. The amount of aluminum nitrate was varied to obtain samples with 2, 4 and 8 atomic percent Al, designated as AZO1, AZO2 and AZO3, respectively. 

Aluminum-doped zinc oxide (AZO) is one of the most important alternatives to ITO and is widely used as a transparent electrode in various kinds of devices. When Al is doped into ZnO, Al^3+^ substitutes Zn^2+^ sites in the ZnO crystal structure resulting in one free electron to contribute to the electric conduction. The ionic radius of Al^3+^ (0.053 nm) is smaller than that of Zn^2+^ (0.074 nm); therefore, Al^3+^ ions can replace Zn^2+^ ions in substitution sites [67]. The electrical conductivity, transparency, thermal stability, and durability make this material attractive [68]. In addition, AZO can also be used as a photocatalyst because of their high activity and chemical stability.

Various methods have been reported for preparing AZO nanopowder [69,70,71,72]. In this study, AZO nanopowders were synthesized by the plasma process using zinc nitrate and aluminum nitrate as the precursors. 

The experimental procedure and the analysis methods were similar to those described above for the synthesis of other oxide nanopowders. The amount of aluminum nitrate was varied to obtain the doping level of 2, 4 and 8 atm % Al designated as AZO1, AZO2 and AZO3, respectively. 

### 4.1. Powder Characteristics

#### 4.1.1. X-ray Diffraction (XRD)

For AZO1 and AZO2, with Al concentration of 2 atm % and 4 atm %, respectively, only ZnO peaks were observed. In sample AZO3 with 8 atm % Al, there were additional peaks observed and were indexed as gahnite phase ZnAl_2_O_4_ (JCPDS 5-0669), implying that the Al content must have exceeded the solubility of Al in ZnO.

The particles are orientated mostly in the direction. Crystallite size was determined from the three peaks (100), (002) and (101) and the average size is calculated. The crystallite size increased with the doping amount and a simultaneous decrease in FWHM was also observed as shown in Table 3. 

#### 4.1.2. Scanning Electron Microscopy—Energy Dispersive Spectroscopy (SEM-EDS)

All plasma-synthesized AZO particles had nearly spherical shapes (Figure 14a), unlike those synthesized by different methods, which ranged from polyhedrals [73], nanorods [74], hexagonal rods [75] and nanowires [76]. Figure 14b shows the EDS spectrum for AZO1 synthesized at 10 kW. It indicates the presence of zinc and aluminum elements where the Zn Lα_1_ and Al Kα_1_ peaks appear at 1.1 and 1.5 keV, respectively.

#### 4.1.3. X-ray Photoelectron Spectroscopy (XPS)

XPS analysis revealed that the oxidation state of Al was close to Al^3+^ in Al_2_O_3_ [77].

#### 4.1.4. Raman Spectroscopy

AZO1 exhibited highly intense Raman peaks suggesting improved crystal quality whereas for AZO3, the reduced intensity of Raman signals indicates the partial crystallinity of AZO nanopowder at higher doping amount [61], which is corroborated by the XRD data. 

### 4.2. Magnetism Measurements of AZO Samples

Magnetic properties of AZO1 and AZO2 were recorded at room temperature and associated M(H) data is shown in Figure 15. The M(H) curves of AZO2 sample clearly indicate the paramagnetic behavior in the high field region, and in the low field region a weak hysteresis is observed. The coexistence of paramagnetism and ferromagnetism has also been reported by other authors [78,79]. AZO1 sample showed a more significant ferromagnetic contribution, evident by its S-shaped curve. The saturation magnetization was attainable in the high field region, which was absent and almost like a linear shape in AZO2. M(H) behavior in the high field region for sample AZO2 was due to the increased occurrence of antiferromagnetic coupling between the Al^3+^ pairs. At a higher doping concentration, the average distance between Al^3+^ ions decrease, resulting in an enhancement of antiferromagnetic contribution. 

The room temperature ferromagnetism mainly arises from two causes: One is the intrinsic magnetism and other is the extrinsic magnetism. Since extrinsic magnetism arises from the formation of secondary phases or the formation of clusters of transition elements [80], it can be ruled out in this case as there was no formation of secondary phases, as observed from the XRD results. The intrinsic magnetism in doped ZnO arises from the exchange interaction between the local spin polarized electrons of Al^3+^ ions and the conductive electrons. Consequently, the polarized conductive electrons undergo an exchange interaction with local spin-polarized electrons of other Al^3+^ ions and thus after successive long exchange interaction, almost all Al^3+^ ions exhibit the same spin direction, resulting in ferromagnetism [81]. The ferromagnetic coupling between the Al dopants and intrinsic defects like oxygen vacancies or zinc interstitials is also responsible for room temperature ferromagnetism [82,83]. It has been reported that Al doping decreases the concentration of oxygen vacancies [84], which is shown by the reaction
Al_2_O_3_ + V_o_^••^ → 2Al^•^_Zn_ + 3O_o_^x^(8)

Thus, the weak ferromagnetism observed in AZO2 is due to a decrease in oxygen vacancies because of Al incorporation in ZnO.

### 4.3. Thin Film Analysis

#### 4.3.1. X-ray Diffraction (XRD) and Scanning Electron Microscopy (SEM)

All the films exhibited a sharp peak at near 35.21°, corresponding to the (002) plane of hexagonal wurtzite structure of ZnO. XRD analysis suggest that that all the AZO thin films exhibited a c-axis orientation perpendicular to the substrate surface. With an increase in Al content, the crystalline quality of AZO thin film improved. The SEM micrographs of thin films revealed that the films had a uniform, dense structure with compact interconnected grains without any signs of porosity.

#### 4.3.2. Electrical Properties

The thickness of all the films measured from the SEM cross-sectional image was 350 ± 5 nm. The I-V curve shows the ohmic behavior. The Al doping mechanism is as follows [85]:2Al_2_O_3_^ZnO^ ⇒ 4Al^•^_Zn_ + 4O_O_^x^ + 2O_2_(g) + 4e^−^(9)

This mechanism suggests that, a decrease in resistivity with Al doping is due to the replacement of Zn^2+^ ions by Al^3+^ ions. Aluminum atoms are incorporated into the ZnO and contribute to the conduction electrons. With an increase in the dopant concentration the resistivity of the thin film was decreased and the minimum resistivity of 9.9 × 10^−4^ Ω·cm was obtained with 4 atm % Al. 

Hall coefficient, carrier concentration and carrier mobility were calculated from the current versus voltage response of the AZO thin films and shown in Table 2. From the Hall measurements, it was observed that, the Hall coefficient was negative, indicating the presence of negative charge carriers. It is clear that with an increase in doping, the carrier concentration increases due to the substitution of more Al^3+^ ions at Zn^2+^ sites of the ZnO structure. 

Annealing of AZO1 thin film was carried out in hydrogen gas at 400 °C for 2 h, which significantly improved the electrical properties of the thin film. The resistivity of AZO1 decreased to 8.68 × 10^−4^ Ωcm after annealing in hydrogen, indicating the generation of free charge carriers. The annealing process can be described by the Kroeger-Vink notation to explain the generation of free charge carriers [85]
2Zn_Zn_^x^ + 2O_O_^x^ + H_2_(g) ⇔ 2Zn_Zn_^x^ + 2OH_O_ + 2e^−^(10)

Another reason for the decrease in resistivity of thin film is that hydrogen annealing removes the adsorbed oxygen, on the surface. Adsorbed oxygen acts as an electron trap and forms a depletion region. Since charge transfer is dominated by the tunneling effect, the large barrier affects the mobility of electrons [86].

Oxygen vacancies formed observed by the PL spectra also acted as an electron donor and permitted the electron movements in the conduction band, thereby improving the conductivity of film. These oxygen vacancies act as electron donors and enhance the transfer of electrons into conduction band leading to enhanced carrier concentration and low resistivity [87].

#### 4.3.3. Optical Properties

The optical transmission spectra of AZO thin films are shown in Figure 16a. The AZO1 and AZO2 thin films exhibited a transmittance of nearly 80% in the visible region whereas the AZO3 film showed a transmittance of 70% in the visible region. 

It was observed that the band gap calculated from Figure 16b increased from 3.2 eV to 3.28 eV with increasing Al concentration, as in AZO2 film. The blue shift exhibited on increasing the Al concentration is associated with the Moss Burstein effect (BM) [88,89]. According to this well-known effect, the conduction band of the degenerate semiconductor is filled with high carrier concentration and the lowest valence energy states are blocked, leading to the lifting of the Fermi level into the conduction band and widening of the optical band gap [31]. Since AZO is an “n” type extrinsic semi-conductor, the Fermi-level gets shifted towards the conduction band in order to conserve the number of particles (mass action law) and to fulfill the overall electrical charge neutrality (neutrality equation) [90]. The results are in good in agreement with the results obtained from the Hall effect measurements where the carrier concentration of AZO2 was higher than that of AZO1 film. 

All the films display a sharp peak of UV near-band edge (NBE) emission as shown in Figure 16c. The crystal quality affects the origin of green emission, hence the improvement of crystal quality (reduction in structural defects such as oxygen vacancies) enhances the near band edge emission with reduction or vanishing of the green emission. The ultraviolet emission is the characteristic emission of ZnO and is attributed to the band edge transitions or the exciton recombination [91,92]. UV near-band edge (NBE) emission is subjected to a blue shift for AZO2 film followed by a red shift for AZO3 film. Blue shift in the UV emission for AZO2 thin film can be attributed to the Burstein-Moss effect. 

## 5. Synthesis of Indium-Doped Zinc Oxide (IZO) Nanopowders

Indium-doped zinc oxide (IZO) nanopowders (wurtzite crystal phase) were synthesized by plasma-assisted chemical vapor synthesis route [93]. Zinc nitrate [Zn(NO_3_)_2_·6H_2_O, Alfa Aesar, Haverhill, MA, USA] powder was the precursor for zinc oxide synthesis, and a mixture of Zn(NO_3_)_2_·6H_2_O and indium nitrate [In(NO_3_)_3_·5H_2_O, Alfa Aesar, Haverhill, MA, USA] precursors was used for the synthesis of indium-doped zinc oxide. 

The amount of indium nitrate was varied to obtain 4 at. % and 8 at. % indium incorporated in zinc oxide, designated as “IZO1” and “IZO2”, respectively. A systematic study was done to analyze the variations in photocatalytic activity and photocurrent properties of IZO with different doping levels. Methylene blue (MB) was used as a model dye for investigating the photocatalytic activity of synthesized nanoparticles.

### 5.1. Characterization

#### 5.1.1. XRD Analysis

The XRD diffraction peaks showed good crystallinity and no In_2_O_3_ or other indium suboxide phases were observed in IZO1. However, in IZO2 sample, the In_2_O_3_ phase was observed along with main wurtzite peaks indicating that at the higher doping amount of 8 atm % indium, excess In^3+^ cannot substitute for Zn sites and was precipitated as indium oxide phases [94]. 

#### 5.1.2. Raman Analysis

It was revealed that highest defect concentrations were present in IZO1, followed by ZnO and IZO2 [95]. 

#### 5.1.3. Band Gap Analysis

Figure 17a shows the UV-visible absorbance spectra of ZnO and In-doped ZnO. With an increase in the indium content, the UV absorption band is shifted to a higher wavelength. IZO1 nanoparticles exhibited maximum absorbance at 406 nm compared to ZnO which exhibited an absorption band at 383 nm. The redshift with an increase in doping amount was also observed in other doped ZnO samples [55]. It was observed in Figure 17b that the band gap of ZnO, IZO1, and IZO2 samples were 3.3 eV, 2.90 eV, and 2.82 eV, respectively. The results indicate that the IZO samples can absorb light not only in the UV but also in the visible light region of the spectrum, suggesting its possibility of being applied as a promising visible light photocatalyst as well.

#### 5.1.4. PL analysis

Spectra showed a strong UV emission peak and a deep-level emission (green emission) peak, as shown in Figure 17c, and the redshift of the UV emission peak was related to the observed variations in band gap [96]. A significant change in the electronic structure of ZnO occurs with indium doping and the intensity of the UV emission peak of the PL spectra gives a direct measurement of the electron-hole recombination and thus high intense spectra demonstrate a high rate of electron-hole recombination [97].

#### 5.1.5. Surface Morphology and Elemental Analysis

AFM images confirmed that at the higher indium doping amount of 8 atm %, the grains become large and some agglomeration was observed contrary to the smaller grain size and even grain distribution with uniform surface morphology of an IZO1 film. This is due to excess indium atoms forming defect phases like In_2_O_3_ phases which segregate to the non-crystalline regions of the grain boundary [98] and can have a deleterious effect on electrical and optical properties in optoelectronic applications [7].

#### 5.1.6. XPS Analysis

The XPS spectra of Zn in ZnO and IZO1 in Figure 18a show two symmetric peaks located at 1021.7 eV and 1044.8 eV corresponding to Zn 2p_3/2_ and Zn 2p_1/2_, respectively. The binding energy difference of 23.10 eV confirmed that Zn exists in +2 oxidized state in IZO. The binding energy peak of Zn 2p_3/2_ spectra in IZO1 was shifted to a higher binder energy value by 0.2 eV compared to the binding energy values of Zn 2p spectra in ZnO. This high binder energy shift is due to the difference in electronegativity values of Zn and In. Since it is more electronegative than Zn, indium picks up electrons from Zn and subsequently decreases the screening effect of electrons for Zn [99]. Peaks observed at 444.3 eV and 452.2 eV shown in Figure 18b correspond to the states of indium 3d_5/2_ and indium 3d_3/2_, respectively, which are representative of the In^3+^ bonding states from In_2_O_3_ [100]. XPS O 1s spectra, shown in Figure 18c, was deconvoluted into three peaks using Gaussian fitting [101]. An increase in the intensity of component O_v_ to O_L_^−2^ component was observed in IZO1 sample compared to the O 1s spectra of ZnO. It has been reported that any changes in the intensity of O_v_ component are connected to a change in the concentration of oxygen vacancies [77]. Thus, indium doping in ZnO influences the concentration of oxygen vacancies.

### 5.2. Photocatalysis Results

The photochemical reactor used was similar to the unit described in Section 3.3 above. The generated electrons and holes in CB and VB upon irradiation with light, act as strong reducing and oxidizing agents and reduce and oxidize many organic and inorganic compounds [102,103]. Among the photocatalysts, ZnO is important for the degradation of organic pollutants which crystallizes in a wurtzite structure and belongs to the space group P6_3_mc [104,105]. ZnO has characteristics of high transparency, good UV trapping properties, non-toxicity, natural abundance, etc., which are important properties of optoelectronic and piezo electronic materials owing to its large band gap of 3.37 eV and a large exciton binding energy of 60 meV [106,107]. However, the major disadvantage in using ZnO nanoparticles is their fast recombination rate of the photogenerated electron-hole pairs, which interferes with the photocatalytic activity of ZnO [108]. In_2_O_3_ is a well-known semiconductor with an indirect band gap of nearly 2.8 eV and it has an excellent electro-optical property together with excellent stability [109]. Still, little is known about their photocatalytic and photocurrent properties. 

On increasing the indium content, the photocatalytic activities increased and IZO1 sample (4 atm %) exhibited the highest activity as shown in Figure 19a,b. On the other hand, with a further increase in indium content as in IZO2 sample (8 atm %), a reduction in the photocatalytic activity was observed. The reaction kinetics of photocatalysis followed a pseudo-first-order kinetics, much like TZO discussed in Section 3. The rate constants were determined to be 0.010 min^−1^, 0.030 min^−1^ and 0.022 min^−1^ for ZnO, IZO1 and IZO2, respectively, as shown in Figure 19c, implying an enhanced photocatalytic activity of IZO1 nanoparticles. Raj et al. [110] reported an increase in the photocatalytic activity of IZO nanoparticles, synthesized by co-precipitation method for Rhodamine-B dye decomposition, and attributed it to defects states and oxygen vacancies. However, no analysis was carried out to determine the variations in defect concentrations. Here, high defect concentrations in IZO1 sample, which are responsible for enhanced photocatalytic activity, were determined from Raman and XPS spectra. Oxygen vacancies promote a reduction in the recombination rate of electrons and holes [111], which was also consistent with the photoluminescence measurements. Photocatalytic activity of IZO2 was lower than that of IZO1 due to formation of defect clusters of In_2_O_3_ phase. This result was also corroborated by XRD analysis. Formed clusters of defect prevent the separation of excited charge carries [57]. Therefore, a nominal content of In^3+^ in ZnO is essential to prevent recombination and also for increasing the lifetime of charge carriers. It is reported that enhanced specific surface area plays an important part in increasing active sites, which increases the concentration of charge carriers [54]. In our study, BET specific surface areas of ZnO, IZO1 and IZO2 samples were measured to be similar in the range of 18–19 m^2^g^−1^and hence specific surface area was not a contributing factor to an enhanced photocatalytic activity observed in IZO1 sample.

## 6. Overall Concluding Remarks

The thermal plasma process performed in this work has shown a tremendous potential as an efficient technique for synthesizing oxide nanopowders. The plasma processing the availability of high temperatures for volatilizing reactants rapidly, followed by vapor phase reactions and rapid quenching to yield nano-sized powder. The plasma process was applied to the synthesis of indium tin oxide (ITO), zinc oxide (ZnO), tin-doped zinc oxide (TZO), aluminum-doped zinc oxide (AZO), and indium-doped zinc oxide (IZO). 

ITO nanopowder thus synthesized was used to prepare thin films, which exhibited high optical transmittance and low electrical resistivity and thus would serve as excellent transparent conducting oxides (TCO). Further, they showed good sensitivity to H_2_ gas around 350 °C. Plasma-synthesized TZO and AZO also displayed properties suitable for optoelectronic applications. The grain size of ITO nanopowder increased with an increase in plasma torch power. Optical transmittance in the visible region approached 85% in ITO1 and ITO2 films. The deposited films showed enhanced electric properties with resistivity in the order of 10^−3^–10^−4^ Ω·cm, and in particular ITO2 showed the best performance: a high carrier concentration of 5.5 × 10^20^ cm^−3^ and a low electrical resistivity of 6.65 × 10^−4^ Ωcm. The ITO1 film exhibited good sensitivity to H_2_ gas, and the sensitivity increased with increases in gas concentration and temperature and reached maximum with 400 ppm of H_2_ gas at an operating temperature of 350 °C.

Results from XRD revealed that the doped and undoped ZnO nanopowders are purely crystalline and belong to the hexagonal wurtzite structure of ZnO. The TZO1 (3 atm % Sn) nano powder exhibited a higher photocatalytic activity than ZnO in the degradation of methylene blue due to increased specific surface area and higher oxygen vacancies. The deposited films exhibited excellent electric properties with resistivities in the order of 10^−3^ Ωcm, TZO1 showed the highest carrier density of 7.6 × 10^19^ cm^−3^ and lowest electrical resistivity of 1.4 × 10^−3^ Ωcm of all films. Optical transmission approached 80% in the visible range for all deposited films, thus making it suitable for optoelectronic applications.

The plasma-synthesized AZO nanopowders were crystalline and of the hexagonal wurtzite structure. Binding energies of Zn 2p_3/2_ and Zn 2p_1/2_ were observed at 1021.42 eV and 1044.34 eV and binding energy of Al 2p core level was observed at 73.8 eV, corresponding to Al^3+^ in Al_2_O_3_. Room temperature ferromagnetism was observed in the AZO1 nanopowder and the contribution of the paramagnetic phase becomes dominant over ferromagnetic phase in the AZO2 nanopowder. AZO films were also prepared on the glass substrates by spin coating using dispersion of nanoparticles. The deposited films showed enhanced electric properties with high carrier concentrations and low resistivity. In particular, AZO2 showed the best performance: a high carrier concentration of 7.7 × 10^20^ cm^−3^ and a low electrical resistivity of 9.90 × 10^−4^ Ωcm. Hydrogen annealing of the AZO1 film enhanced the electrical properties with a low resistivity of 8.7 × 10^−4^ Ω·cm and a mobility value of 32 cm^2^/V·s. Optical transmittance in the visible region approached 80% in the AZO1 and AZO2 films. 

The IZO1 (~4 atm % indium) nanopowder synthesized by plasma showed enhanced photocatalytic activity than ZnO and IZO2 due to the low rate of recombination rate of electrons and hole evidenced from the PL spectra and the higher defect concentrations as revealed from the Raman spectra. A detailed mechanism for the enhanced photocatalytic activity of the IZO nanoparticles was proposed and it was determined that the main oxidative species involved in the degradation of MB were h^+^ and O_2_^•−^. The deposited IZO1 film showed enhanced photoelectric properties that were consistent with the enhanced photocatalytic activity in IZO1. The present novel plasma-assisted chemical vapor synthesis route supplied new perspectives on the bandgap engineering of IZO and also shed light on the influence of indium doping on the recombination rate of photogenerated charge carriers. 

## Figures and Tables

**Figure 1 molecules-26-01456-f001:**
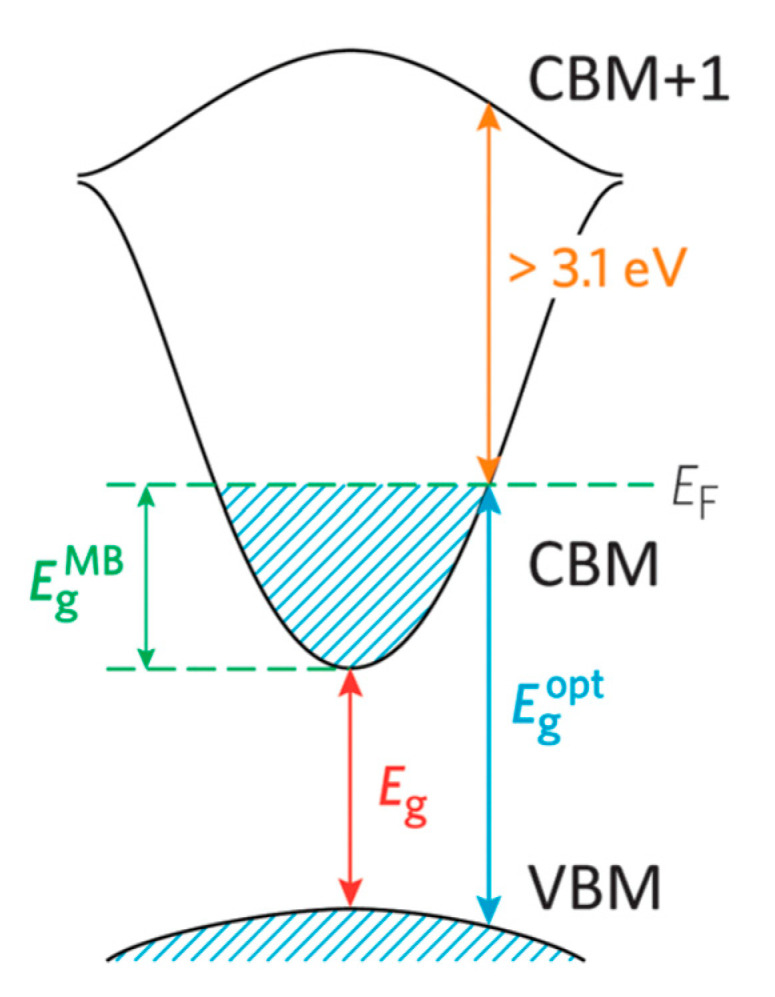
Diagram of the optical widening by the effect of the Moss–Burstein shift. Reproduced from Ref. [6], an open access article.

**Figure 2 molecules-26-01456-f002:**
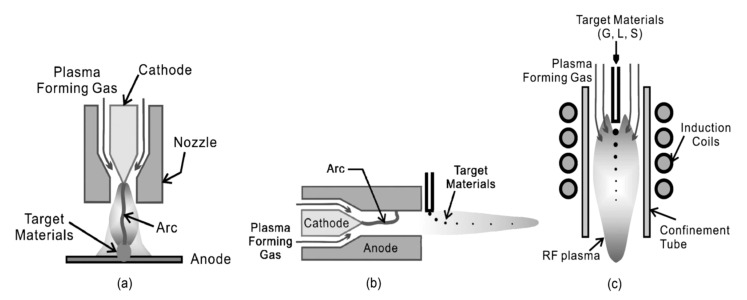
Schematic Diagrams of the Typical Thermal Plasma Torches Available for Synthesis of Nano-sized Powders. (**a**)Transferred DC Plasma Torch, (**b**) Non-transferred DC Plasma Torch and (**c**) RF Plasma Torch. Reproduced from Ref. [11], an open access article.

**Figure 3 molecules-26-01456-f003:**
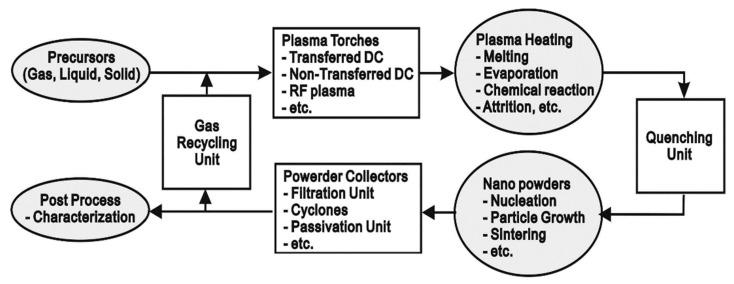
Thermal Plasma Synthesis Procedure for the Production of Nano-sized Powders. Reproduced from Ref. [11], an open access article.

**Figure 4 molecules-26-01456-f004:**
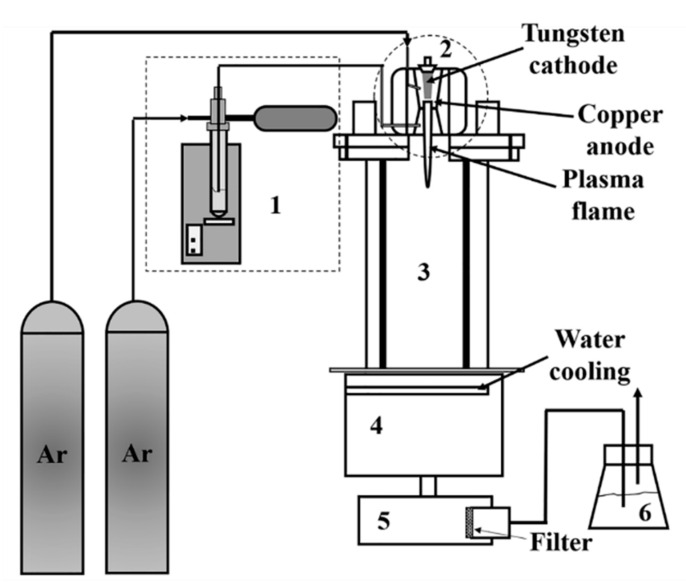
Schematic diagram of plasma reactor system: (1) powder feeding system, (2) plasma gun, (3) reactor chamber, (4) cooling chamber, (5) powder collector, and (6) scrubber. Reproduced with permission from Ref. [15]; published by IOP Publishing, 2018.

**Figure 5 molecules-26-01456-f005:**
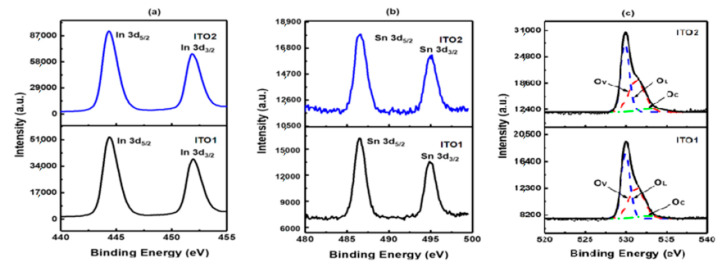
(**a**) XPS In 3d narrow spectra of samples: ITO1 and ITO2; (**b**) XPS Sn 3d narrow spectra of samples: ITO1 and ITO2; (**c**) XPS O 1s narrow spectra of samples: ITO1 and ITO2. Reproduced with permission from Ref. [15]; published by IOP Publishing, 2018.

**Figure 6 molecules-26-01456-f006:**
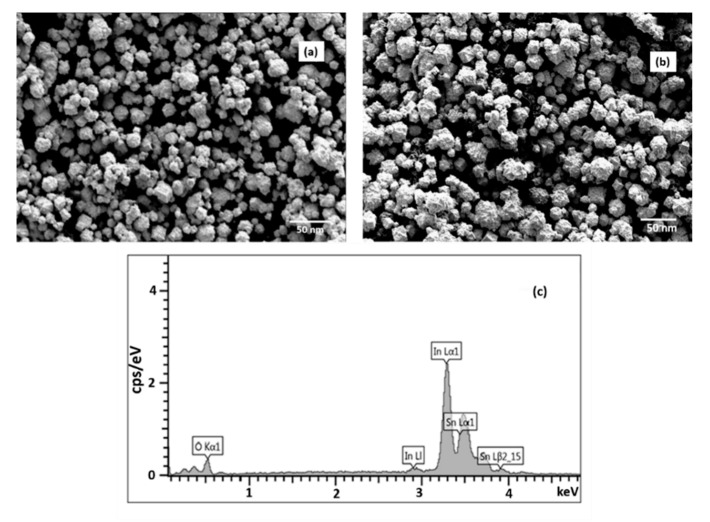
(**a**) SEM micrographs of ITO1 nanopowder synthesized at 10 kW, (**b**) SEM micrographs of ITO1 nanopowder synthesized at 20 kW, (**c**) Energy-dispersive X-ray Spectrum of ITO1 nanopowder synthesized at 10 kW. Reproduced with permission from Ref. [15]; published by IOP Publishing, 2018.

**Figure 7 molecules-26-01456-f007:**
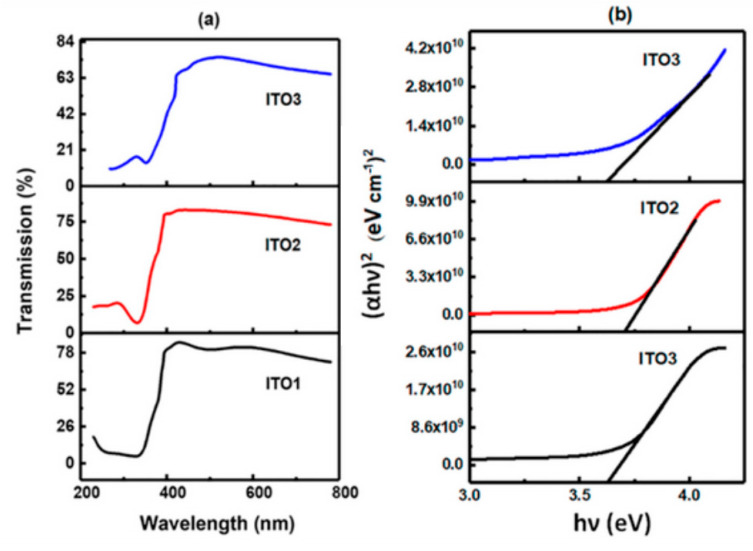
(**a**) Transmission curves of ITO films, (**b**) Tauc plots to determine the band gap. Reproduced with permission from Ref. [15]; published by IOP Publishing, 2018.

**Figure 8 molecules-26-01456-f008:**
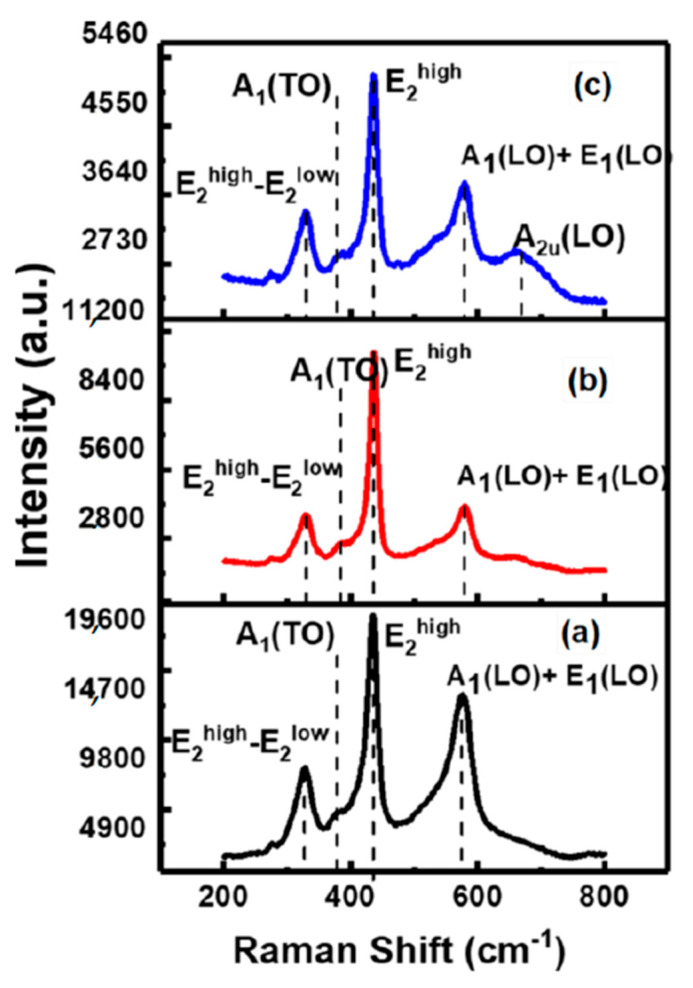
Raman spectra of (**a**) ZnO, (**b**) TZO1 and (**c**) TZO2. Reproduced with permission from Ref. [32]; published by Springer Nature, 2018.

**Figure 9 molecules-26-01456-f009:**
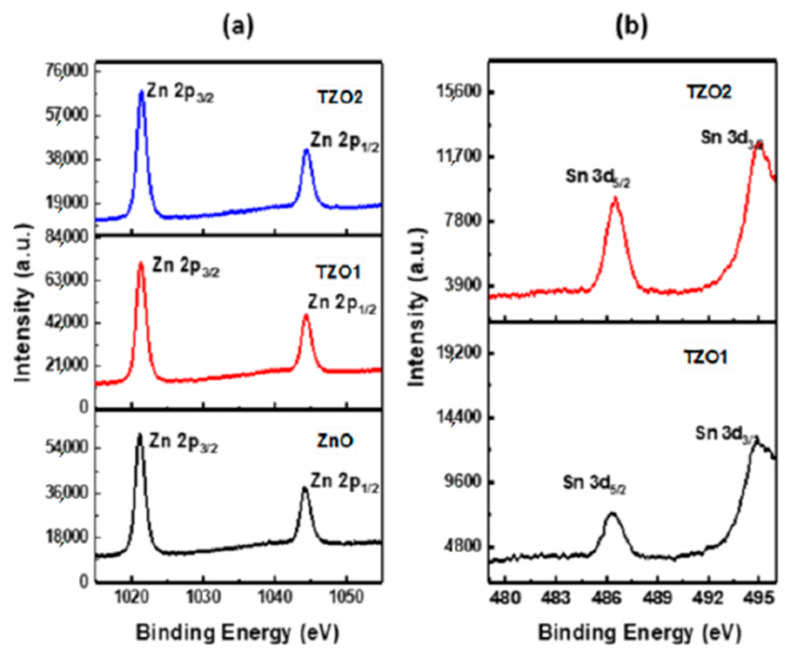
(**a**) XPS Zn 2p core level spectra of TZO1 and TZO2. (**b**) XPS Sn 3d spectra of TZO1 and TZO2. Reproduced with permission from Ref. [32]; published by Springer Nature, 2018.

**Figure 10 molecules-26-01456-f010:**
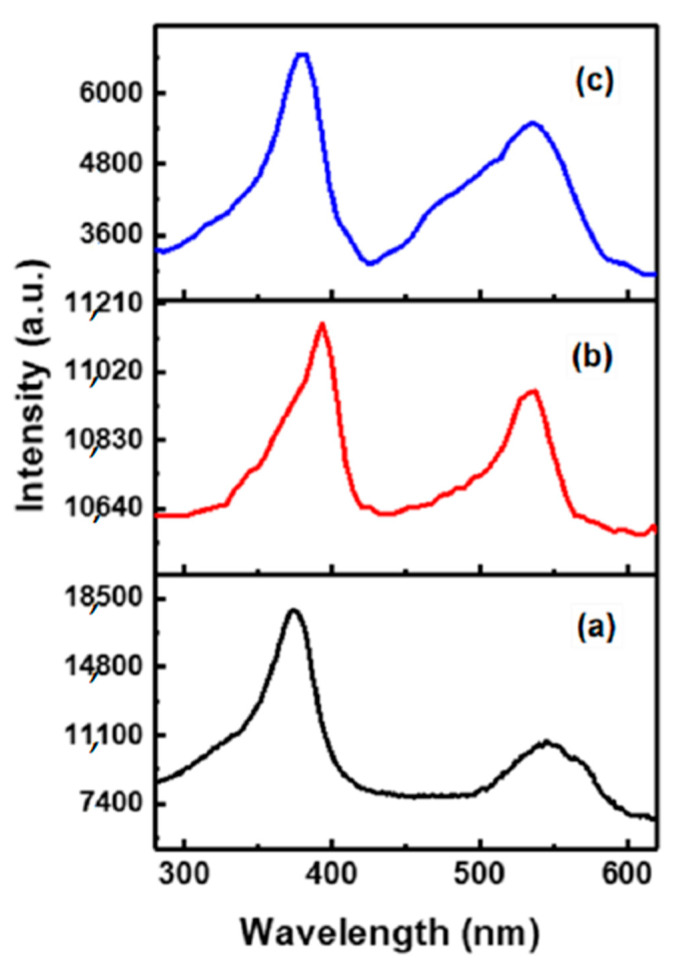
PL spectra for (**a**) ZnO, (**b**) TZO1, and (**c**) TZO2. Reproduced with permission from Ref. [32]; published by Springer Nature, 2018.

**Figure 11 molecules-26-01456-f011:**
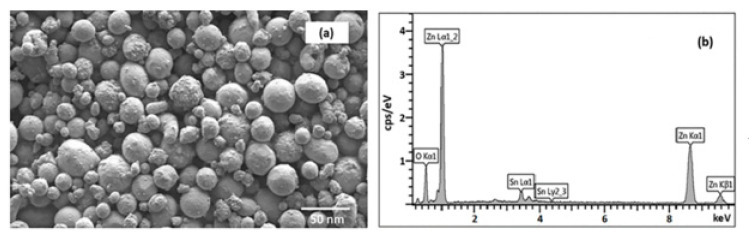
(**a**) SEM micrographs of TZO1 particles synthesized at 10 kW. (**b**) EDS spectrum for TZO1 sample synthesized at 10 kW. Reproduced with permission from Ref. [32]; published by Springer Nature, 2018.

**Figure 12 molecules-26-01456-f012:**
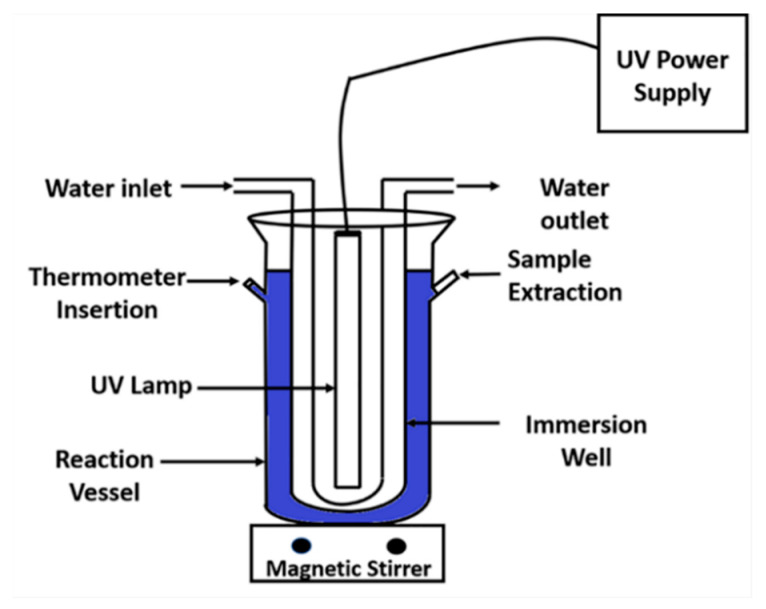
Schematic diagram of the experimental set-up used for photocatalysis tests. Reproduced with permission from Ref. [32]; published by Springer Nature, 2018.

**Figure 13 molecules-26-01456-f013:**
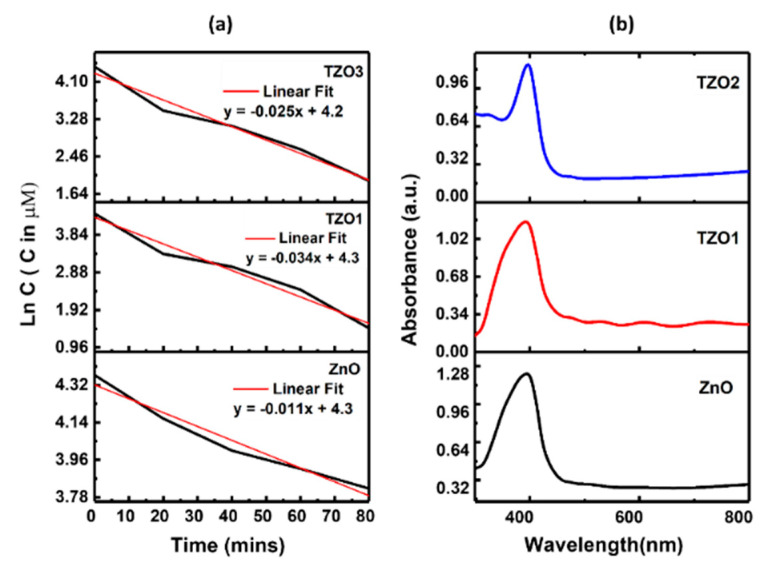
(**a**) Plot showing the linear regression curve fit for the natural logarithm of absorbance of the methylene blue concentration against irradiation time for ZnO, TZO1 and TZO2. (**b)** Absorbance spectrum of ZnO, TZO1 and TZO2. Reproduced with permission from Ref. [32]; published by Springer Nature, 2018.

**Figure 14 molecules-26-01456-f014:**
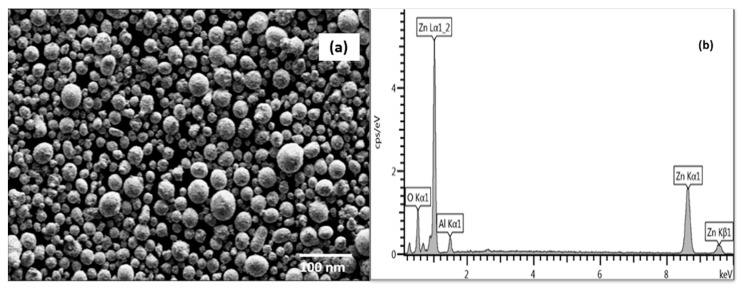
(**a**) SEM micrographs of a) AZO1 particles synthesized at 10 kW. Spherically shaped morphology with no cluster formation is observed for all the compositions of AZO samples. (**b**) EDS spectrum for AZO1 synthesized at 10 kW. Reproduced with permission from Ref. [61]; published by Springer Nature, 2019.

**Figure 15 molecules-26-01456-f015:**
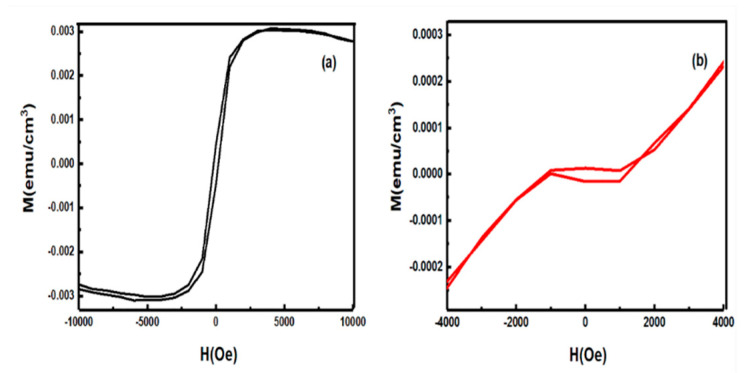
M(H) data for (**a**) AZO1 and (**b**) AZO2. Reproduced with permission from Ref. [61]; published by Springer Nature, 2019.

**Figure 16 molecules-26-01456-f016:**
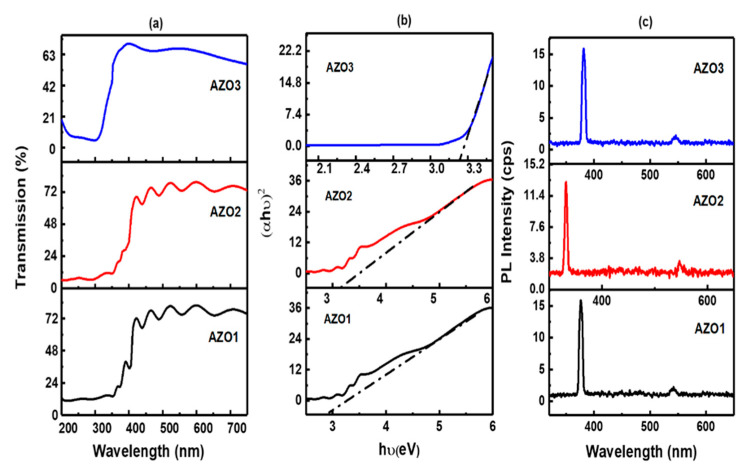
(**a**) Transmission curves for AZO1, AZO2 and AZO3. (**b**) Tauc plots to determine the band gap for AZO1, AZO2 and AZO3. (**c**) PL spectra for a) AZO1, b) AZO2 and c) AZO3. Reproduced with permission from Ref. [61]; published by Springer Nature, 2019.

**Figure 17 molecules-26-01456-f017:**
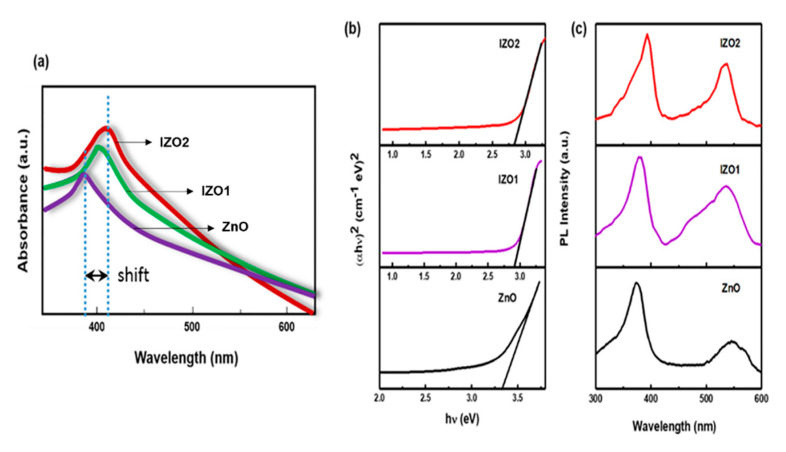
Optical characterization results of IZO samples: (**a**) Absorbance spectra of ZnO, IZO1 and IZO2. (**b**) Tauc’s plot of (*αhν*)^2^ versus energy hν. (**c**) PL spectra of ZnO, IZO1 and IZO2. Reproduced with permission from Ref. [93]; published by Elsevier, 2019.

**Figure 18 molecules-26-01456-f018:**
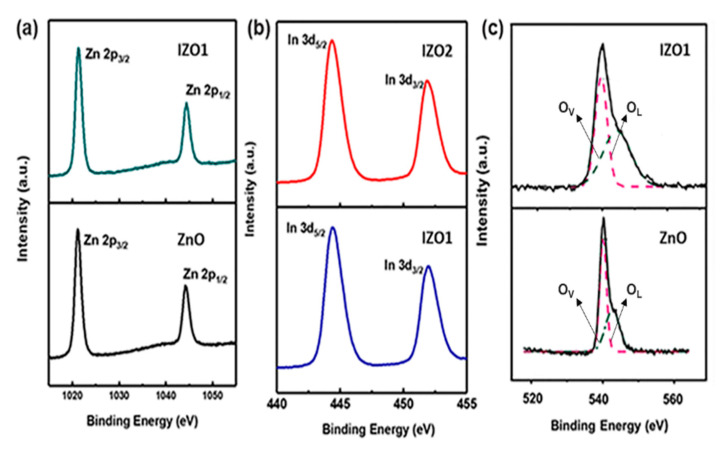
XPS spectra of ZnO and IZO films (**a**) XPS Zn 2p core level spectra of ZnO and IZO1. (**b**) XPS In 3d spectra of IZO1 and IZO2. (**c**) XPS O1s spectra for (**a**) ZnO and (**b**) IZO2. Reproduced with permission from Ref. [93]; published by Elsevier, 2019.

**Figure 19 molecules-26-01456-f019:**
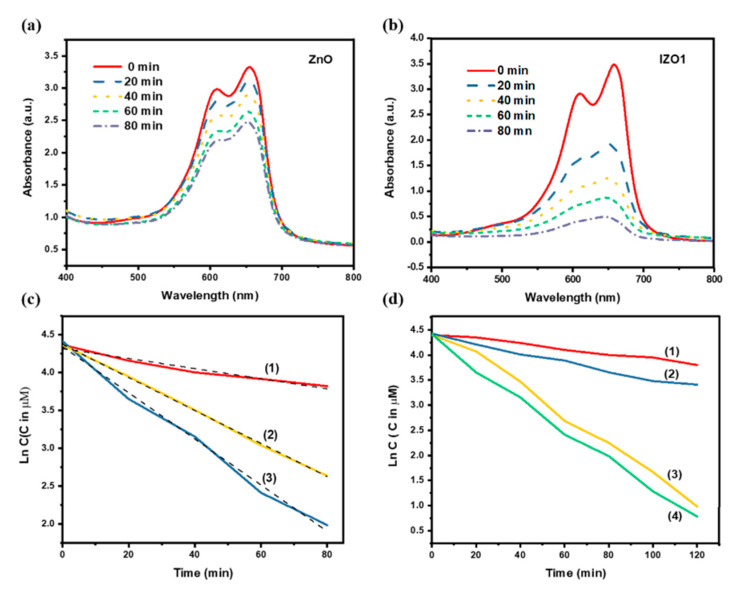
Absorption properties and photodegradation tests. Absorbance spectra of methylene blue at regular intervals of time using photocatalysts (**a**) ZnO and (**b**) IZO1. (**c**) Degradation rate constants obtained by plotting Ln C vs. time for (1) ZnO (2) IZO2 and (3) IZO1. (**d**) Effect of addition of different scavengers (1) addition of 1 mM EDTA (2) addition of 1 mM p-BQ (3) addition of 1 mM *t*-BuOH and (4) no scavenger addition towards photodegradation of methylene blue using IZO1 photocatalyst. Reproduced with permission from Ref. [93]; published by Elsevier, 2019.

**Table 1 molecules-26-01456-t001:** Crystallite size calculated from XRD analysis. Adapted from Ref. [32].

Type	Size (100) nm	Size (002) nm	Size (101) nm	Average Crystallite Size (nm)	Average Lattice Constant “a” (Å)
ZnO	56.3	59.4	56.2	57.3	3.28
TZO1	21.4	24.7	20.2	22.1	3.27
TZO2	38.5	38.4	38.3	38.4	3.27

**Table 2 molecules-26-01456-t002:** The electrical properties of ZnO and TZO films. Adapted from Refs. [15,32,61].

Type	Type of Film	Carrier Density (cm^−3^)	Mobility (cm^2^/V·s)	Resistivity (Ωcm)
ITO	ITO1	1.8 × 10^20^	14.0	2.47 × 10^−3^
	ITO2	5.5 × 10^20^	17.1	6.65 × 10^−4^
	ITO3	2.2 × 10^20^	3.64	7.79 × 10^−3^
TZO	ZnO	6.6 × 10^19^	11.6	8.2 × 10^−3^
	TZO1	7.6 × 10^19^	58.7	1.4 × 10^−3^
	TZO2	1.8 × 10^20^	15.1	2.3 × 10^−3^
AZO	AZO1	2.2 × 10^20^	23.7	1.2 × 10^−3^
	AZO2	7.7 × 10^20^	8.2	9.9 × 10^−4^
	AZO3	6.7 × 10^20^	5.5	1.7 × 10^−3^
	AZO1(Annealed)	2.2 × 10^20^	32.1	8.7 × 10^−4^

**Table 3 molecules-26-01456-t003:** Crystallite size calculated from XRD analysis. Adapted from Ref. [61].

Type	Size (100) nm	Size (002) nm	Size (101) nm	Average Crystallite Size (nm)	Average Lattice constant “a” (Å)
AZO1	29.4	28.5	28.4	28.8	3.28
AZO2	33.4	37.2	37.4	36.0	3.26
AZO3	39.4	41.7	41.2	40.7	3.26

## Data Availability

The data presented in this study are available on request. The data are not included in this article due to the limitation of space.

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
