# Peer review of "Plasma Synthesis of Advanced Metal Oxide Nanoparticles and Their Applications as Transparent Conducting Oxide Thin Films"

_molecules, 2021, doi:10.3390/molecules26051456_

Round 1

Reviewer 1 Report

According to manuscript 1118312, entitled „Plasma Synthesis of Advanced Metal Oxide Nanoparticles“ by authors H. Y. Sohn and A. Murali The presented manucript is an extensive review of deposition and research of metal oxides materials in nanopowder and in thin film states. The plasma assisted chemical vapor syntensis has been applied for metal oxide nanopowder deposition. The studied materials are ITO (nanopowder and thin films), ZnO, tin doped ZnO and Al doped ZnO (nanopowder and thin films) and indium doped zinc oxide nanopowder. The used characterization techniques reveal their structural, optical properties. Different metal oxides are studied in respect to their photocatalytic activity (ZnO, Sn doped ZnO, In doped ZnO), electrical properties (Sn doped ZnO films), magnetic properties (Al doped ZnO). The authors used a concise manner of expression. The paper is acceptable for publication after some minor revision: 1. The paper is a review of extensive research and some data is given in brief. It will be good to cite the corresponding papers, where the authors present the detailed studies and discussions. For example, the XRD and Raman analysis of In doped ZnO. .

Reviewer 2 Report

The authors must decide if the title will be kept containing oxide nanoparticles or oxide nanocoatings.

Section 1.2 seems excessive as it doesn't contribute with a new information. If not completely removed, at least it should be shortened.

There is no sufficient critical analysis of the results, just mixing of previous studies without smooth interconnection between the parts.

More appropriate comparative tables are needed to visualize the general picture of the coatings overviewed. The tables are now split into smaller ones located in the subsection, but this doesn't give crucial comparative information about the positive and negative features of the coatings.

All summary sub-sections must be removed and their content is better to be moved as a general conclusion.

Adhesion and sheet resistance of the coatings are not commented. Transmission of 80% for TZO is rather low according to the current standard. You should better highlight why we need to synthesize them if their transparency is not satisfactory.

Page 18 and p.19 mentioned SEM study, but SEM images are not provided.

In general no uniform style of information presenting is followed - different analyses are provided for the different materials.

Reviewer 3 Report

The authors summarized recent works in their laboratory on the synthesis of advanced transparent conducting oxide nanopowders by the use of plasma. This review has the characteristics of clear logic, breakthrough point and strong innovation. Therefore, this manuscript can be accepted after minor revision.

Strangely, this manuscript is more of a research paper than a review, and figures from self-published articles are also required to be quoted and authorized. Of course, if these data are not yet published, this manuscript would qualify as a research paper, which shows that the authors have done a comprehensive job in this area. But as a review, it needs to be a brainstorming, comprehensive summary of the published work of other scholars, not just the work of the author's team.

In addition, the literatures summarized by the authors are somewhat out of date, and the work of the past two years has not been included, which needs to be improved.

Round 2

Reviewer 2 Report

Dear Authors,

I consider your response suitable and I recommend acceptance of the new version.